# MEMORY-MODULAR CLASSIFICATION: LEARNING TO GENERALIZE WITH MEMORY REPLACEMENT

## ABSTRACT

We introduce a memory-modular learner for image classification that externalizes knowledge memorization from reasoning and thus effectively generalizes to new classes by replacing memory contents. Instead of statically compiling the world knowledge and task skills into model weights during training, the proposed model stores the knowledge in an external memory of image/text data and learns to dynamically select relevant contents from the memory according to an input image. The meta-learned model performs robust classification with a memory of noisy web-crawled data and adapts to new classes without re-training when the memory is replaced. Experimental results show the promising performance of our method on diverse scenarios, including zero-shot/few-shot classification of unseen classes, fine-grained classification, and class-incremental classification.

## 1 INTRODUCTION

Large-scale neural models have been achieving remarkable results when fine-tuned and applied to downstream tasks in computer vision (Kolesnikov et al., 2020; Yuan et al., 2021; Alayrac et al., 2022) and natural language processing (Brown et al., 2020; Touvron et al., 2023). With a massive scale of data and compute for training, they typically compile the world knowledge and task skills into a huge number of entangled model parameters. This makes it difficult to interpret which knowledge in training data or learned skill causes the models to produce a specific output, and, more importantly, the models are not able to immediately reflect the change in world knowledge *i.e.*, updated data sources for target tasks, without additional training.

Recent work on retrieval-augmented models (Guu et al., 2020; Hu et al., 2023) attempts to decouple knowledge memory from reasoning. It enables the models to use external knowledge sources and better dedicate the model parameters to reasoning. However, these models are limited to specific knowledge-intensive tasks, such as question answering (Gao et al., 2022) or long-tailed classification (Long et al., 2022), and focus on improving performance rather than investigating the generalization ability of learned models with severe memory update or replacement.

In this work, we introduce a *memory-modular learner* for image classification that performs adaptive reasoning using an external memory that is replaceable and modular. In particular, we focus on *memory-modular generalization* to unseen classes, *i.e.*, how effectively the learner generalizes to new classes simply by replacing the contents of the memory without additional training. The memory-modular classifier stores image/text data in knowledge memory and meta-learns to dynamically select relevant contents from the memory according to an input image. The proposed model can classify an input image into target classes that are represented by class names, text descriptions, or support images. It also performs robust classification with a memory of noisy web-crawled data and immediately adapts to new classes when the memory is replaced.

One key observation of our work is that by modeling classification as a *metric-learning* (Snell et al., 2017), our model overfits less to the actual content of the memory and learns more about how to reason with any given memory content. More specifically, we represent class weights as *class prototypes*, which are generated from memory items, rather than learnable parameters. This design choice allows us to easily update the memory and adapt to new classes without having to relearn the entire model. Thanks to its flexibility, our meta-learned model can handle zero to multi-shot samples, as well as a variable number of classes according to updated memory. Experimental results demon-

strate the promising performance of our method on diverse scenarios, including zero-shot/few-shot classification of unseen classes, fine-grained classification, and class-incremental classification.

Our contributions can be summarized as follows.

- We present a memory-modular learner for image classification that performs adaptive reasoning using an external replaceable memory.
- We provide in-depth analyses on the memory-modular generalization to unseen classes in a realistic setup with web-crawled noisy memory.
- We show that our method naturally applies to various scenarios, achieving promising performance in zero-shot, few-shot, fine-grained, and class-incremental classification.

## 2 RELATED WORK

### 2.1 LEARNING TO GENERALIZE WITH FEW-SHOT DATA

Few-shot image classification (Fei-Fei et al., 2006) aims at meta-learning to generalize beyond the predefined classes seen in training, given a few images from arbitrary target classes. Conventional experimental setup of few-shot classification Vinyals et al. (2016); Triantafillou et al. (2020) assumes at least a few hundred shots during (meta-)training before the actual few-shot inference stage, here we take a different angle of employing a generalist pre-trained encoder and leveraging a few labeled examples, *e.g.*, $\leq 16$, in both for (meta-)training and testing. Following the convention, the term "shot" in this manuscript is used to denote the number of given class-annotated images for each target class. Zero-shot classification takes one step further than few-shot classification in terms of reducing label dependence; classification is conducted based on textual information of the images, *e.g.*, yes-or-no attributes (Lampert et al., 2013) or the class name in text (Socher et al., 2013), of arbitrary classes with zero image example. A recent work of a vision-and-language foundation model (Radford et al., 2021) tackles zero-shot classification by learning from 400 million image-and-text paired similarity via cross-modal contrastive learning, which co-embeds the multi-modal features on a single embedding space and enables zero-shot inference based on cross-modal similarity. Leveraging the power of this generalist feature encoder, we make our model dynamically accesses to a memory, which stores world knowledge from the internet, to obtain the relevant knowledge to the input. The obtained external knowledge is integrated in feature augmentation, of which related work is continued in the following.

### 2.2 MEMORY-AUGMENTED IMAGE RECOGNITION

Employing external sources as a memory for classification has a long and rich history, *e.g.*, Hart (1968) introduces the nearest neighbor (NN) classifier that retrieves NN of the input and performs majority voting for class prediction. Khandelwal et al. (2020); Nakata et al. (2022) modernize the $k$NN classifier with the help of pre-trained neural models. The proposed simple and learning-free method revisits the potential of memory-based reasoning apart from knowledge encoding (Graves et al., 2014), however, the use of laboriously annotated images of target classes restricts real-world applications. The memory-based learning methods such as Jia et al. (2021); Long et al. (2022); Iscen et al. (2023b) propose methods that access the memory and use the retrieval results in image recognition, where the memory consists of image-text pairs. On the other hand, our approach assumes a more weakly-supervised type of memory of separate image and text memory, assuming no individual cross-modal correspondence labeled in advance. The work of Wei et al. (2023); Hu et al. (2023) leverage web-crawled image-text pairs to train a powerful image-text feature encoder, which takes an orthogonal angle to our approach that takes a frozen image-text encoder as a multi-modal feature extractor. Leveraging memory has been adopted for zero-shot classification in the work of Iscen et al. (2023a), whereas we present a versatile memory-based method for zero-shot classification as well as for three different classification tasks.

## 3 MEMORY-MODULAR LEARNER

We address the problem of classifying an input image into target classes that are represented by a class name or text description, *i.e.*, zero-shot classification, or additional few support images,

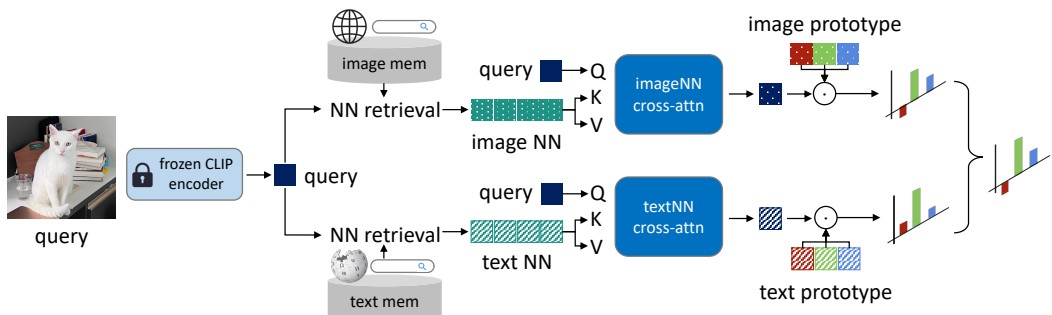

Figure 1: Overview of the memory-modular learner

*i.e.*, few-shot classification. To this end, we introduce a *memory-modular learner* that performs adaptive reasoning using an external memory that is updatable and replaceable. In particular, we focus on *memory-modular generalization* to unseen classes, *i.e.*, how effectively the learner generalizes to new classes simply by replacing the contents of the memory without additional training. Our memory-modular learner leverages two modalities of vision and language using the pre-trained CLIP encoder (Radford et al., 2021) as a base feature extractor for image and text. Since our method is not specific to CLIP, any other image-text model, *e.g.* LiT (Zhai et al., 2022), can be adopted instead. Figure 1 illustrates the overall architecture of our approach.

The memory-modular classifier starts by loading the knowledge memory and generating class prototypes for target classes (Section 3.1). These memory items and prototypes are all stored as features in the embedding space of a pre-trained image-text model. They are replaceable whenever the target classes or the external knowledge sources are updated. Given an input image, the memory-modular classifier accesses the knowledge memory, retrieves relevant items, and predicts the corresponding class by multi-modal interaction with class prototypes (Section 3.2). Since the class prototypes are generated based on the memory items, the prototype-based classifier can handle new target classes of updated memory contents without additional training.

## 3.1 MEMORY AND PROTOTYPE CONSTRUCTION

Given the target class names or descriptions, we construct the knowledge memory based on image and text data and generate class prototypes using the image-text memory. Note that all the memory items can be added or deleted, and even completely replaced without affecting the model weights as the world knowledge is updated.

**Knowledge memory.** The image memory is constructed based on images collected from the web with simple keyword search. For each target class $c$, images are collected using the class name $c$ as the search keyword on Google and Flickr. These web-crawled images and texts may be noisy but consist in scalable memory contents that reflect the world knowledge. We follow a similar strategy for the text memory. We query Wikipedia with each target class $c$, and collect the relevant textual information.

After collecting the relevant images and texts for each target class $c$, we extract their features with the image-text encoder, and then store them in the image and text memory: $\mathcal{M}_c^{\text{img}} = \{\boldsymbol{v}_i\}_{i=1}^{N_c^{\text{img}}}$ and $\mathcal{M}_c^{\text{txt}} = \{\boldsymbol{t}_j\}_{j=1}^{N_c^{\text{txt}}}$, respectively.

**Class prototypes.** We construct class prototypes based on cross-modal consensus between image and text memory items. For each target class $c$, we first compute cross-modal average similarity from each image item to all text items, and then select the top-$K$ images in terms of the average image-to-text similarity. The image prototype for class $c$ is then set to be their average feature:

$$\boldsymbol{p}_c^{\text{img}} = \frac{1}{|\mathcal{T}|} \sum_{\boldsymbol{v} \in \mathcal{T}} \boldsymbol{v}, \qquad \mathcal{T} = \operatorname{argmax}_{\boldsymbol{v}' \in \mathcal{M}_c^{\text{img}}}^K \left( \sum_{\boldsymbol{t} \in \mathcal{M}_c^{\text{txt}}} \frac{\boldsymbol{v}' \cdot \boldsymbol{t}}{|\boldsymbol{v}'||\boldsymbol{t}|} \right), \tag{1}$$

where $\text{argmax}_{\boldsymbol{s} \in \mathcal{S}}^{K}(\cdot)$ represents the top-$K$ operator that returns the best $K$ elements from the set $\mathcal{S}$ maximizing a target function. This cross-modal process allows us to obtain effective class prototypes without any supervision. The text prototype is obtained using the average text-to-image similarity in a similar manner.

When additional support samples, *e.g.*, images or texts, are available for each class, we produce class prototypes simply using the support samples instead of the top $K$ images in Equation 1. In this way, our model naturally extends to such scenarios with additional samples, *i.e.*, few-shot classification, which is also validated in experiments.

## 3.2 Reasoning with memory access

Given an input image for classification, we incorporate memory knowledge into the reasoning. Relevant items are retrieved from image and text memory, and the retrieved items are aggregated by cross-attention. The resultant image and text features are integrated into the input feature, which is then correlated with the class prototype of the corresponding modality. The predictions from the two branches are eventually merged at the logit level, producing class predictions.

**Memory retrieval.** For an input image feature $\boldsymbol{f}$ extracted from the image encoder, its $k$-nearest-neighbor image items are selected based on cosine similarity from the knowledge memory of all target classes:

$$\mathcal{N}^{\text{img}} = \text{argmax}_{\boldsymbol{v} \in \cup_c \mathcal{M}_c^{\text{img}}}^{k} \left( \frac{\boldsymbol{f} \cdot \boldsymbol{v}}{|\boldsymbol{f}||\boldsymbol{v}|} \right). \tag{2}$$

The $k$-nearest-neighbor text elements are also retrieved in a similar manner by querying the input image feature to the text memory.

**Attentive knowledge integration.** The knowledge of the retrieved memory items $\mathcal{N}^{\text{img}} = [\boldsymbol{v}_i]_{i=1}^{k}$ is aggregated by cross-attention and then integrated with the input embedding $\boldsymbol{f}$, which learns the interaction between the input and the retrieved items:

$$\boldsymbol{f}^{\text{img}} = \boldsymbol{f} + \sigma \left( \frac{\mathbf{Q}(\boldsymbol{f}) \cdot [\mathbf{K}(\boldsymbol{v}_i)]_{i=1}^{k}}{\sqrt{d}} \right) [\mathbf{V}(\boldsymbol{v}_i)]_{i=1}^{k}, \tag{3}$$

where $\mathbf{Q}, \mathbf{K}, \mathbf{V}$ are projection layers with non-linearity and $[\cdot]$ is concatenation. Likewise, the same step with the text $k$NN features is conducted in parallel and produces $\boldsymbol{f}^{\text{txt}}$. This process can be viewed as a learnable soft integration of memory knowledge in contrast to conventional hard voting in memory-based classification, *e.g.*, Nakata et al. (2022).

**Multi-modal inference.** The resultant embedding is matched against the multi-modal prototypes $\boldsymbol{P}^{\text{img}} = [\boldsymbol{p}_c^{\text{img}}]_{c=1}^{C}$ and $\boldsymbol{P}^{\text{txt}} = [\boldsymbol{p}_c^{\text{txt}}]_{c=1}^{C}$ for all $C$ target classes to produce classification scores as

$$\boldsymbol{z} = \frac{\boldsymbol{P}^{\text{txt}} \cdot \boldsymbol{f}^{\text{txt}}}{|\boldsymbol{P}^{\text{txt}}||\boldsymbol{f}^{\text{txt}}|} + \frac{\boldsymbol{P}^{\text{img}} \cdot \boldsymbol{f}^{\text{img}}}{|\boldsymbol{P}^{\text{img}}||\boldsymbol{f}^{\text{img}}|}, \quad \boldsymbol{z} \in \mathbb{R}^C. \tag{4}$$

The final class prediction is then made simply by taking the class with the highest score.

## 3.3 Training

Our model is trained with cross-entropy loss with one-hot ground-truth class label $\boldsymbol{y}$ and the logit:

$$\mathcal{L} = -\sum_{c=1}^{C} \boldsymbol{y}_c \log \frac{\exp(\boldsymbol{z}_c/\tau)}{\sum_{c'}^{C} \exp(\boldsymbol{z}_{c'}/\tau)}, \tag{5}$$

where $\tau$ is a temperature for scaling. Note that we freeze the pre-trained image-text encoder and train the remaining part only, corresponding to a single attention layer on the image and text branches. This helps retain general knowledge in the pre-trained encoder, avoid overfitting, and reuse pre-computed features of memory items. In our case, the number of training parameters amounts to 6.3M in total. This small part effectively learns memory-based reasoning for classification, as will be demonstrated in the experimental section.

## 4 EXPERIMENTS

In this section, we first report experimental results on zero-shot classification along with ablative studies on the model design choices. We also provide experimental observation and analyses, as well as validation on various classification setups such as few-shot classification, fine-grained classification, and class-incremental learning. We will make our code and data publicly available.

### 4.1 EXPERIMENTAL SETUP

**Training details.** For the feature extractor, we use the pretrained CLIP-B/32 and CLIP-L/14. For training, we use a batch size of 256, a learning rate of $1e^{-6}$ and weight decay of $5e^{-4}$ on a single 2080 Ti or an RTX 3090 GPU for all training and testing. We retrieve 32 NNs for both the image and text memory. We use $K = 16$ for prototype construction and set the learning temperature $\tau = 16$, which are chosen via hyperparameter search on validation sets. We use three random seeds for training for drawing random few-shot support sets and report the average.

**Baseline methods.** For zero-shot classification baselines, we employ the $k$NN classifier (Nakata et al., 2022)[1] and zero-shot CLIP. **$k$NN classifier** retrieves $k$NN of the input and immediately predicts the class by majority voting. **Zero-shot CLIP** represents each class as the class name in a text template, *e.g.*, a photo of [class], and matches it with the input query. For few-shot classification baselines, we employ a state-of-the-art few-shot classifier (Hu et al., 2022), which leverages a foundation model encoder and fine-tunes the model weights. **Linear prob** is another few-shot classification baseline, where we simply add a fixed-length linear layer on top of the frozen CLIP backbone and train it with the target class few-shot examples. We also compare ours with a memory-based recognition work, Retrieval-Augmented Classification (**RAC**) (Long et al., 2022). RAC first retrieves the nearest images from an image memory and feeds their corresponding class text labels to a subsequent text encoder to obtain an auxiliary textual feature, which is added to the input image feature. RAC was originally for long-tailed classification, thus we reproduce it by training with online few-shot examples. During reproduction, we have to freeze the text encoder, otherwise, the learning fails with few-shot training inputs.

**Evaluation datasets.** For zero-shot/few-shot, fine-grained, and class incremental learning, we use available public benchmarks and/or establish a new benchmark by splitting classes within an existing dataset. For **zero-shot classification**, we adopt CUB-200-2011 (Wah et al., 2011) of 200 bird species, following the public class split from (Akata et al., 2013) of 150/50 classes for training/validation. Unlike the convention, we use only 5 labeled samples from training out of the average 50 samples, which we call CUB-F (Few training samples). Likewise, we introduce an ImageNet (Russakovsky et al., 2015) split in a similar manner such that it comprises of 600/200/200 classes for train/validation/test, and call it ImageNet-F. For **few-shot classification**, a standard few-shot image classification benchmark called mini-ImageNet (Vinyals et al., 2016) is used, which consists of 64/16/20 object classes for train/validation/test splits, respectively. We reuse CUB-F and ImageNet-F splits for few-shot classification simply by allowing a learner to use a few labeled samples (shots) from validation and test set. For **fine-grained classification**, we employ Food-101 (Bossard et al., 2014) of 101 food types and CUB again, but with its full 200 classes. For **class incremental classification**, we adopt a scenario called ImageNet100-Base0-Inc10 (Rebuffi et al., 2017), where 10 classes are incrementally added for 10 stages, resulting in the accumulation of 100 classes evaluated at the end. To construct the external image memory for ImageNet derivatives, we employ a publicly available web-crawled image dataset, Webvision (Li et al., 2017) of which contents are crawled with the 1000 classes of ImageNet1K from Google and Flickr. We use the image subset crawled from Google unless otherwise specified. To construct image memory for fine-grained classification, as no public web-crawled datasets for the corresponding classes are available, we actually crawl maximum 500 images for each class text labels from Google with an auto crawler. For text memory, we crawl text sentences in a Wikipedia article titled with the class name for all the classes. In such way, the modest length of memory is obtained, *e.g.*, 0.7M images and 0.2M texts corresponding to the 1K classes of ImageNet1K, of which $k$NN search is implementable with a regular PyTorch (Paszke et al., 2017) `topK` module.

---

[1]As it assumes to access a fully-annotated dataset for simulating image memory, which is considered impractical, we replace it with the noisy web-crawled memory.

Table 1: Zero-shot classification

| method | ImageNet-F | | CUB-F | |
|---|---|---|---|---|
| backbone | B/32 | L/14 | B/32 | L/14 |
| Zero-shot CLIP | 82.8 | 90.1 | 61.9 | 73.8 |
| $k$NN classifier | 80.6 | 86.5 | 67.9 | 82.6 |
| Yu et al. (2020) | - | - | 72.4 | |
| Chen et al. (2022) | - | - | 76.1 | |
| Ours | 83.0 | 91.1 | 75.6 | 87.8 |

Table 2: Results on memory replacement

| WebVision (WV) source as memory | | ImageNet-F |
|---|---|---|
| train&val | test | |
| WV-Google | WV-Google | 83.0 |
| | WV-Flickr | 83.1 (+0.1) |
| WV-Flickr | WV-Flickr | 83.2 |
| | WV-Google | 83.2 (-0.0) |

Table 3: Retrieval and classification performance

| task → | retrieval | | cls. |
|---|---|---|---|
| memory | recall@1 | recall@16 | acc. |
| ImageNet1K (clean) | 66.4 | 93.5 | 84.7 |
| Webvision (noisy) | 65.5 | 90.3 | 83.0 |

Table 4: Accuracy in varying $k$ of $k$NN

| $k$ of $k$NN | 8 | 16 | 32 | 64 |
|---|---|---|---|---|
| $k$NN classifier | 75.2 | 76.3 | 76.4 | 76.0 |
| Ours | 83.0 | 83.1 | 83.0 | 82.7 |

## 4.2 ZERO-SHOT CLASSIFICATION

Table 1 presents the comparison between our model and other zero-shot models. Among them, Yu et al. (2020); Chen et al. (2022) are state-of-the-art zero-shot learners, which are trained with *all the annotated training images and text attributes* in the dataset on top of an ImageNet-trained ResNet101 (He et al., 2016). In contrast, our model uses only a few labeled examples from the training split (of the non-target classes), and no annotated attributes are used during training. In this setup, the only information given to represent a target class at inference is a phrased class label, *e.g.*, "tabby cat", which is used as the search keyword in web searches for memory construction. As shown in the results, our model consistently outperforms the others. In particular, comparing with $k$NN classifier baseline, which is uni-modal and non-learnable, our method demonstrates the effectiveness of meta-learning to integrate $k$NN knowledge in multi-modalities. The following paragraphs provide analyses and ablation studies on zero-shot classification.

**Robustness to memory replacement.** Table 2 shows the results of memory replacement where we replace memory contents once a zero-shot memory-modular learner is trained. Note that the zero-shot model already interacts with a completely unseen memory of unseen target classes in the experiments of Table 1. The experiment in Table 2 further examines whether the model can perform robustly to different memory contents, yet of the same set of target classes. To this end, we use two disjoint sets of images equipped in WebVision (WV) for image memory; one set is obtained through Google crawling, while the other is from Flickr with both class sets sharing the 200 target classes. Once the model is trained with one memory source, we replace it with the other from the two different memory sources for testing. The results show that the model retains the classification accuracy even when replacing the test memory. This suggests that the proposed model generalizes beyond certain contents of memory.

**Memory retrieval performance.** Table 3 shows the memory retrieval quality and the end task performance by comparing clean- and noisy-labeled memory datasets, ImageNet1K and Webvision, respectively. In contrast to the web-crawled memory, ImageNet provides the same domain of images of the target query with cleaner class labels than Webvision. The recall@$k$ is an instance retrieval metric that returns 1 if any instances from the ground-truth class are included in the $k$NN and 0 otherwise (Jégou et al., 2011). The ImageNet1K memory is considered as an upperbound in terms of the data noise. Comparing the retrieval and classification performance, the cleaner the memory is, the more accurate $k$NN items are retrieved, translating to the higher end task accuracy when the $k$NN knowledge is integrated through learning.

**Performance with varying $k$ of $k$NN.** We vary $k$ for two retrieval-based models, the $k$NN majority voting classifier (Nakata et al., 2022) and our memory-modular learner, and compare their accuracy in Table 4. Unlike the $k$NN majority voting classifier, our model learns to perform soft aggregation of the $k$NN values. Looking at the result, both the $k$NN retrieval models present a sweet spot at certain $k$, in this example at around 16. Our memory-modular learner shows outperforming performance than the hard majority voting classifier, which is not learnable.

Table 5: Comparison with different memory types

| kNN retrieval from | ImageNet-F | | CUB-F | |
|---|---|---|---|---|
| | zero-shot | few-shot | zero-shot | few-shot |
| No kNN | 75.6 | 73.5 | 60.0 | 61.0 |
| Text memory | 82.3 | 81.7 | 69.4 | 69.8 |
| Image memory | 76.2 | 78.4 | 71.2 | 70.3 |
| Unified memory | 76.8 | 78.6 | 71.3 | 70.4 |
| Ours (separete memory) | 83.0 | 83.9 | 75.6 | 76.9 |

Table 6: Comparison with different class prototypes

| prototype | ImageNet-F | CUB-F |
|---|---|---|
| Avg mem | 81.2 | 74.4 |
| Text only | 82.7 | 69.1 |
| Image only | 82.3 | 76.8 |
| Ours (image & text) | 83.0 | 75.6 |

Figure 2: Examples of the retrieved image and text kNNs. Human faces are anonymized.

**Comparison with different memory types.** To validate our separate memory design, we ablate the memory components and compare the results in Table 5. The "no kNN" baseline has the same architecture with the complete model, but instead, it takes the input feature for the key and value inputs in replace of the kNN. This no kNN baseline results in the lowest performance and signifies the importance of the kNN knowledge integration. Next, we ablate either image or text memory. It is noteworthy that the model with only the image memory is more effective than using the text memory on CUB-F and the reverse for ImageNet-F, suggesting that the diverse visual aspects collected from the internet provides the useful features for fine-grained visual classification, while the textual information is useful on coarse-grained classification of general objects. Lastly, we merge the VL memory into a single one, retrieve the modality-agnostic kNN features, and then pass them to a single knowledge integration branch. We observe that the majority of the nearest neighbors are retrieved from the image memory, thus reaching closely to the image-memory model performance. To effectively interact with multi-modal NNs, we choose to separate the VL memories as well as the reasoning branches. This memory ablation results signify that the two-branch multi-modal memory retrieval plays an essential role in terms of performance.

**Comparison with different class prototypes.** Table 6 compares different methods to build class prototypes. We first try to naïvely average each memory to build prototypes without using the top $K$ operator in Equation 1, which poorly performs on both the datasets. We also attempted to share the prototype from either memory for the two knowledge integration branches. We observe that the image prototype contributes higher than the text one does on CUB-F and the reverse on ImageNet-F, similarly observed in Table 5, suggesting that the efficacy of the image and text prototype can be dependent on target dataset characteristics. Our method, which uses online multi-modal prototype, generally achieves robust performance on both the datasets, as the prototype contains the condensed multi-modal knowledge extracted from the VL memory with outlier filtering (Equation 1) based on VL consensus. Note that the learnable linear classifier is not considered in class prototype options as it is not able to adapt to the arbitrary or unseen classes without additional training.

Table 7: Few-shot classification

| method | mem | ImgNet-F
16 shots | mini-ImgNet
1 shot |
|---|---|---|---|
| Linear-prob CLIP | | 80.6 | 61.3 |
| RAC | ✓ | 78.1 | 69.1 |
| Hu et al. (2022) | | - | 95.3 |
| Ours | ✓ | 84.3 | 96.6 |

Table 8: Classification results on seen classes with few-shot training images

| method | mem | ImgNet1K
16 shots | CUB200
5 shots | Food101
5 shots |
|---|---|---|---|---|
| Zero-shot CLIP | | 63.4 | 51.8 | 68.3 |
| Linear prob CLIP | | 62.0 | 56.2 | 69.2 |
| $k$NN-fewshots | | 48.2 | 38.9 | 52.4 |
| $k$NN-memory | ✓ | 56.0 | 49.2 | 70.2 |
| RAC | ✓ | 62.2 | 56.8 | 73.1 |
| Ours | ✓ | 68.4 | 65.6 | 83.4 |

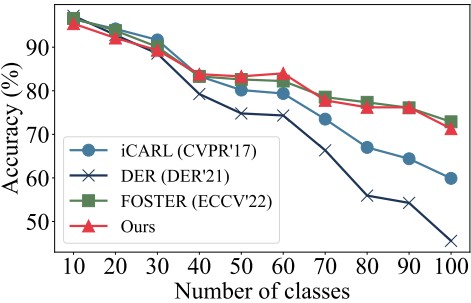

Figure 3: Class-incremental learning

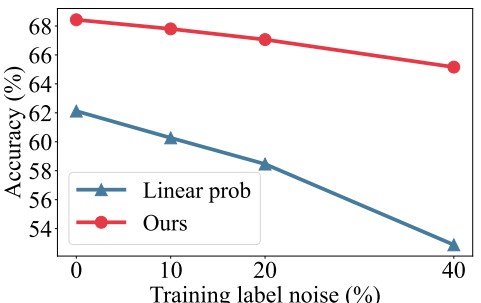

Figure 4: Comparison with training label noise

**Visualization of $k$NN.** Figure 2 visualizes the retrieved two image NNs and two text NNs of the given input query along with the corresponding prediction of zero-shot CLIP at the side. Note that the images and texts in the example are independently retrieved from each memory. From image retrieval, we observe that noticeable patterns of the input query tend to appear on the retrieved image NNs. From text, we observe that retrieved texts often contains synonymous keywords, *e.g.*, the scientific names of animals, or visually describes the given image, *e.g.*, "sharp-tailed" in the second query example. The last line demonstrates a query image containing multiple semantic objects and thus ambiguously labeled, where the model retrieves semantically similar NNs to the query.

### 4.3 APPLICATION TO OTHER CLASSIFICATION SETUPS

Here we provide observations and analyses through extensive ablation studies as well as evaluation results on few-shot and class incremental learning setups.

**Few-shot classification.** In addition to zero-shot classification, the few-shot classification setup allows for additional few-shot samples for each target class at inference. In this experiment, we consider the support examples are images, *i.e.*, few-shot image classification. Table 7 compares our model and others on few-shot classification. We experience that RAC, a memory-based feature learning method, is hard to train with a few shots and underperforms than others. Our model is also validated to be effective on few-shot classification.

**Supervised image classification.** In contrast to unseen class generalization, the more primitive and standard definition of image classification assumes that the given training set and evaluation set share the target class set. In other words, the exploring the classes beyond training is not the focus of this standard classification setup. Our memory-modular learner can also tackle this classification by setting the memory and the prototype not changed between training and testing. The results on such experiment are shown in Table 8. The two baseline models "$k$NN-fewshots" and "$k$NN-memory" are $k$NN classifiers that retrieves $k$NNs from the given training few-shot samples or the web-crawled image memory, respectively. Even the labeled examples from target classes are used, the $k$NN-fewshots baseline achieves the lowest accuracy, which implies the amount of the image memory for retrieval should be large enough for robust retrieval. The proposed model, learning to aggregate the web-crawled knowledge, signifies its effectiveness also on the standard classification, and noticeably, the improvement is more significant on the fine-grained datasets.

Table 9: Comparison with increasing shots from target classes on ImageNet-F

| methods | test-time training | 4 shots | 16 shots | 64 shots | 128 shots | 256 shots |
|---|---|---|---|---|---|---|
| (a) Linear prob | ✓ | 72.0 | 80.6 | 85.5 | 86.9 | 87.3 |
| (b) Ours | | 82.8 | 83.5 | 85.7 | 85.7 | 85.8 |
| (c) Ours* | ✓ | 83.4 | 87.0 | 87.4 | 87.9 | 88.3 |
| (b) - (a) gap | | +10.7 | +2.9 | +0.2 | -1.2 | -1.4 |
| (c) - (a) gap | | +11.3 | +6.4 | +1.9 | +1.0 | +1.0 |

**Class-incremental learning.** We compare class-incremental learning methods and ours on a standard class-incremental learning benchmark on ImageNet100-Base0-Inc10. The primary focus of class-incremental learning is not to forget the previously seen classes with limited access to them, while a majority of the work such as iCARL, DER, FOSTER (Rebuffi et al., 2017; Yan et al., 2021; Wang et al., 2022) uses a memory to store seen-class data. Unlike our model, their usage of memory is rather re-training the model. For a fair comparison, we reproduce the three methods with the CLIP-B/32 backbone as baseline, and also implement our model with following the standard evaluation constraints: the memory is given as the in-domain class images of the new classes of the current stage and the total memory length is always fixed as the number of classes increases. Our method works on par with the recent work as drawn in Figure 3 as the number of classes increases. Note that ours competes with them without using specific techniques for this specific task such as distillation of old class knowledge in model weights or storing the heavy model weights to the memory.

**Training label noise robustness.** We showcase that the reasoning procedure via memory retrieval is robust against the training data label noise. To simulate this label noise, we randomly permute from 10% to 40% of the class labels of training queries with something else and train the architecture with the corrupted labels. This comparison validates the effectiveness of reasoning source for classification: reasoning from the semantically relavent $k$NN contents *vs.* reasoning from the memorized parameters without such external knowledge. Figure 4 presents the comparison of the baseline and ours each of which is trained with the increasing portion of incorrect class labels on ImageNet1K. Our memory model achieves consistently higher accuracy, furthermore, the performance gap between the baseline also increases with more portion of incorrect labels. We suspect that the reasoning via memory access encourages robust learning against to the training label noise as the model collaborates with the multi-modal $k$NN features, providing a collaborative reasoning with the embedding neighborhood.

**Performance with increasing shots.** The previous experiments focus on leveraging memory knowledge with minimal supervision for target classes. Table 9 shows the performance when increasing the number of shots to a larger number, compared to additional test-time training. While the linear prob (a) is directly trained with $M$ shots for each target class, our method (b) is trained on the non-target classes and tested *without* additional training, *i.e.*, target information. We observe that the memory-modular model outperforms the baseline by a significant margin with 4 shots, and the gain gradually diminishes as the target-specific additional training is done with more shots. The results of our method (c) show that the same additional training on our model recovers the diminished gap and even further improves the performance. This demonstrates that additional test-time training benefits from the memory-modular meta-learning.

## 5 CONCLUSION

We have presented the memory-modular classifier and demonstrated its efficacy on various scenarios, investigating the memory-modular generalization for unseen classes. The experiments show that our memory-modular reasoning exhibits robustness to noisy data in the memory and the replacement of memory contents. Although the memory access introduces space and time overhead than conventional models, the overhead is greatly reduced thanks to the memory design; the memory consists of pre-computed features of the frozen encoder, which are reused for inference except for memory replacement. We believe that memory-modular learning benefits various tasks in the areas of artificial intelligence beyond classification, leaving them for future work.

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
