# OpenReview forum: "Memory-Modular Classification: Learning to Generalize with Memory Replacement"
_ICLR.cc/2024/Conference — ICLR 2024 Conference Withdrawn Submission_

### Official Review · Reviewer_3JpF · 2023-10-27

**Soundness:** 3 good
**Presentation:** 3 good
**Contribution:** 2 fair
**Rating:** 3
**Confidence:** 3

**Summary:**

This paper introduce a memory-modular learner for image classification that externalizes
knowledge memorization from reasoning and thus effectively generalizes to new
classes by replacing memory contents.  Experimental results show the promising performance of their method
on diverse scenarios.


In conclusion, this paper develops a memory module for images and text, which can be seen as an augmentation method.
In my opinion, this method is simple and useful, but maybe not suitable for ICLR, because it does not provide a new sight or novel model.
The work is more likely an incremental work and I advised the author choose another conferance, such as CVPR.

**Strengths:**

1) the idea  is simple and useful;

2) the experiments show the method is effective on many different datasets.

**Weaknesses:**

In my opinion, the work is incremental, and I dont find any novel module in this work apart from experiments.

**Questions:**

1. please talk about detailed difference between your work and other memory augment multi-models.ork.

---

### Official Review · Reviewer_BjqH · 2023-10-30

**Soundness:** 3 good
**Presentation:** 3 good
**Contribution:** 3 good
**Rating:** 6
**Confidence:** 5

**Summary:**

The paper introduces an external memory of texts and images from the Internet onto a fixed CLIP image encoder and meta-trains two cross-attention modules for image aggregation and text aggregation, respectively. After meta-training, the resulting model can be applied to a variety of classification tasks including general, few-shot and class incremental classification.

**Strengths:**

The paper is well written.
The method is simple and effective.
The model seems flexible in terms of manipulating memory components.
The effectiveness of the proposed method is evaluated through many experiments, supporting the conclusions.

**Weaknesses:**

1) Lack of some implementation details. For example, after modifying either the text memory or image memory, will the image and text prototypes also be updated? The current description of the class-incremental learning experiment is not clear to me. Does the text memory and image memory only contain the exemplars from the current stage? Then how does the model perform well under class incremental classification as the whole system relies on the outputs of two aggregation modules to make predictions? Does that mean that the output features from the contextual modules are largely similar to the query features since one can regard this as CLIP zero-shot features which do not change with modified representations?

2) Lack of discussion of related works in continual learning, especially a recent work that similarly contains an external memory [1]. It might make sense to include a subsection in the related work.

3) Additional experiments of the aggregation module should be included. The current implementation uses cross-attention. It might make sense to compare to some baselines including self-attention (including the current query and the retrievals as a sequence of tokens to a self-attention block), linear attention (at the cost of fixed k), non-trainable cross-attention (without mapping linear layers), etc. These experiments help to fortify the current design choices.

4) Experiments in Tab. 2 can be enhanced through changing text sources as well. I would be intrigued to see the outcome.

[1] Zhen Zhu et al. Continual Learning in Open-vocabulary Classification with Complementary Memory Systems.

**Questions:**

Please see 1) in the weaknesses.

**Details Of Ethics Concerns:**

The paper uses crawler to crawl data from the Internet which might cause issue of copyright.

---

### Official Review · Reviewer_ELpJ · 2023-11-02

**Soundness:** 2 fair
**Presentation:** 2 fair
**Contribution:** 2 fair
**Rating:** 5
**Confidence:** 4

**Summary:**

This paper introduces a memory-modular learner for image classification that performs adaptive reasoning using an external replaceable memory. The proposed model can generalize to unseen classes simply by replacing the memory contents without re-training. The experimental results show that the proposed method can apply to various scenarios, including zero-shot, few-shot, fine-grained, and class-incremental classification.

**Strengths:**

+ The paper provides a versatile memory-based method for different classification tasks due to representing class weights as class prototypes that are generated form memory items.

+ The proposed memory-modular learner has good generalization ability to unseen classes by replacing or updating the contents of the memory.

**Weaknesses:**

- On the whole, the table arrangement is very irregular and cannot clearly express the superiority of the results of the proposed method. To make matters worse, the table data contains errors, such as Table 9. For the figures: for the sake of standardization, the texts in Figure 1should be unified into “Times New Roman” format; The pictures in Figure 2 are arranged misaligned; In Figure 3, it can use “cite/citep” to cite the corresponding article, which is more formal.
- In Section 3, the proposed framework of learner looks similar to the [1] (reduced version). Are there any specific differences between them?
[1] Ziniu Hu et al. Reveal: Retrieval-augmented visual-language pre-training with
multi-source multimodal knowledge memory, CVPR 2023.

- Eq.1 exists errors. First, T calculates the average similarity rather than sum of features. Secondly, the image and text features should be explicitly stated whether L1 norm or L2 norm is used, and represented with the format “||a||” (the following equations are the same). Thirdly, some characters in the equation are not explained, such as ‘σ,d' in Eq.3.

- In the experiments part, please provide the detailed analysis to the experimental results. Specifically, (1) In the zero-shot classification experiments (Table 1), why the results of Chen et al. method are higher than the proposed method on CUB-F? (2) In the memory replacement experiments, from the results of Table 2, it seems that replacing the memory content has no impact on the test results. Can you provide other datasets for testing? And provide detailed result analysis to illustrate the generalization performance. (3) In varying k experiments (Table 4), it does not indicate which dataset is used for this result. In addition, from the results, when k<32, the performance of the proposed method is not greatly affected; when k > 32, the performance drops significantly. Please provide a detailed explanation. (4) Table 7, ‘mini-ImgNet’ should indicate which dataset abbreviation it is in the text or caption. (5) In the class-incremental experiments, is the backbone used (i.e., CLIP-B/32) in each method is pre-trained or need to be re-train? Generally speaking, the results of DER are much better than of iCarL, but why is it opposite in Figure 3. And can you provide the results of using the Resnet model as backbone?

**Questions:**

See my comments in Weaknesses.

---

### Official Review · Reviewer_sn6x · 2023-11-02

**Soundness:** 3 good
**Presentation:** 2 fair
**Contribution:** 3 good
**Rating:** 5
**Confidence:** 4

**Summary:**

This paper presents a memory-modular learner for image classification, which dynamically selects appropriate information from memory containing previously learned knowledge. This study can contribute to zero-shot/few-shot learning.

**Strengths:**

1. Learning of images and texts together provides an effective setting for zero/few shot learning.
2. Experiment results show significant improvement.
3. Modular approach is effective for better generalization and continual learning.

**Weaknesses:**

1. Paper writing needs some improvement, e.g., please fix "we make our model dynamically
accesses to a memory"
2. In few-shot setting, please compare with more SOTA methods.

**Questions:**

1. In zero shot setting, competing methods "Yu et al. (2020); Chen et al. (2022) are state-of-the-art zero-shot learners, which are trained with all the annotated training images and text attributes in the dataset on top of an ImageNet-trained ResNet101 (He et al., 2016)." while the proposed study: "The image memory is constructed based on images collected from the web with simple keyword search. For each target class c, images are collected using the class name c as the search keyword on Google and Flickr. These web-crawled images and texts may be noisy but consist in scalable memory contents that reflect the world knowledge. We follow a similar strategy for the text memory. We query Wikipedia with each target class c, and collect the relevant textual information." Is this a fair comparison? Is it possible that these crawled images/texts actually contain images in testing set?